# Study of Bag Grouting Pile Reinforcing Deep Soft-Soil Foundation with an Interlayer of Hard Materials on High-Speed Railway Ballast Track

**Shen Zuo [1], Qingyu Zhang [1], Jin Li [1,*], Peng Liu [2,*], Fengkun Cui [1] and Ying Chen [3]**

[1] School of Transportation Civil Engineering, Shandong Jiaotong University, Jinan 250357, China; 204096@sdjtu.edu.cn (S.Z.); 20107011@stu.sdjtu.edu.cn (Q.Z.); 204118@sdjtu.edu.cn (F.C.)

[2] School of Civil Engineering, Central South University, Changsha 410004, China

[3] School of Civil Engineering, Central South University of Forestry and Technology, 498 Shaoshan Road, Changsha 410004, China; t20192437@csuft.edu.cn

\* Correspondence: 204026@sdjtu.edu.cn (J.L.); liupeng868@csu.edu.cn (P.L.)

**Abstract:** High-speed railways are built in deep soft-soil foundations interlayered with hard materials. However, the hard layers cannot be penetrated by conventional foundation treatment. Moreover, the project cost is sometimes prohibitive, which is an issue for post-construction subsidence control. The bag grouting pile is a special grouting-pile-reinforcement technique for treating soft-soil foundations interlayered with hard materials. This paper conducted tests on the Ningbo-Taizhou-Wenzhou ballast track passenger railway, combining a conventional mixing-pile-foundation (not through hard layer) processing station and the bag-grouting-pile-processing subgrade line. Using field tests combined with numerical calculation, the lateral displacement, settlement and pile–soil stress were tested to obtain the working properties of the grouting-pile-composite foundation. The variation in the law of lateral displacement, settlement, and pile–soil stress of the foundation was studied. The results showed that the post-construction subsidence of bag-grouting-pile- and conventional mixing-pile-foundation control was less than 15 cm and 30 cm, respectively, which met the design requirements. The results also showed that the design scheme was reasonable (load-sharing ratio of bag-pile-foundation pile soil-based test was reasonable). The numerical calculation showed a change in pile diameter. It also showed that pile spacing could not improve subsidence-control properties, bag grouting pile can promote pore-pressure dissipation and accelerate consolidation, and pile spacing can be increased to cut project costs incurred by the pile-bearing capacity.

**Keywords:** bag grouting pile; deep soft soil; interlayer; hard materials; field test; finite-element calculation

## 1. Introduction

The east coast of the Zhejiang and Fujian provinces has large deposits of silt and silty soil, as much as 30 to 45 m thick and 70 m high in some areas. The soil is characterized by high water content (40–70%), high compressibility, a porosity ratio (the ratio of the volume of the pores to the total volume) of 1.0 to 2.0, and a compression coefficient that ranges from 0.5 to 2.0 MPa$^{-1}$. Some areas have a large amount of gravel or hard plastic clay interlayer and lens distributed in the soil. It is difficult and expensive to break through the composite foundation of a conventional pile to reach the hard layer, especially when using a bored pile or replacing the road with a bridge. Therefore, researchers have developed a special reinforcement technology for deep soft-soil foundations interlayered with hard materials [1]: the bag-grouting-pile method, which was used to reinforce the foundation of the Ningbo-Taizhou-Wenzhou Railway. There are nearly ten reports on its design and construction technology, including design calculation [2], construction technology [3], subsidence characteristics [4], and engineering applications [5,6]. Wu, Z. used the ultrasonic-transmission method and the inclinometer method to detect the pile-diameter change in the

bag grouting pile; the small-strain test was used to detect the pile-body integrity of the bag grouting pile [7]. Li, Q. used the inclinometer tube to detect the horizontal displacement of the soil around the pile when the pile was formed and obtained the change trend of the excess pore water pressure during the pile-formation process and the maximum value of the horizontal displacement of the soil around the pile on the upper and lower sides [8]. Wang, D. obtained the settlement deformation and distribution law of bagged piles through layered settlement pipes and inclinometer pipes [9]. T. N. Lohani obtained the distribution of vertical compressive strength and transverse shear strength of bagged piles through a full-scale loading test [10]. The test showed that bagged piles have strong anisotropic strength characteristics. There are few measured data on site. There is also a lack of research results that systematically analyze the working behavior of bagged-pile-composite foundations, predict post-construction settlement, and discuss the effect of reinforcement.

The bag grouting pile, which is a new type of composite foundation developed by the author, was applied to the Ningbo-Taizhou-Wenzhou ballast track passenger railway for the first time. However, its reinforcement effect remains to be verified. A situ test was conducted in order to clarify the effect of reinforcement. The working properties of the bagged-pile-composite foundation were obtained by testing parameters such as lateral displacement, settlement, pile–soil stress, etc. Additionally, numerical calculation was used to analyze the influence of different design parameters on the reinforcement effect. The research results of this paper are the experimental data of the first application of bag grouting piles in a high-speed railway. The data can be used to reveal the action mechanism of the bag grouting pile and verify its reinforcement effect; it is helpful for the popularization and application of the foundation-treatment technology of the bag grouting pile, and has important engineering significance for the development of railway construction.

## 2. Test Background

The Ningbo-Taizhou-Wenzhou Railway is located in the southeast coastal area, where the soil is hard-layered deep soft soil. We obtained the following rock strata after drilling the soil: (1) clay, sallow, soft plastic; (2) silt, grey, plastic flow; (3)-1 silty clay, sallow, hard plastic; (3)-2 fine pebble soil, grayish-purple to light yellow, slightly dense to medium dense; (4) muddy clay, gray, fluid plastic; (5)-1 fine pebble soil, grayish-purple to grayish-yellow, slightly to medium dense; (5)-2 silty clay, gray, soft plastic; (5)-3 clay, grayish-white to grayish-yellow, soft plastic; (6) clay, gray and yellow, soft plastic; (7) fine gravel soil, gray and yellow, slightly~ medium dense; (8) tuff, weathered, III level.

At 0–9 m deep, the soil is made up of clay and silt; at 8–15, it is made up of gravelly soil—the hard layer—and at 15 m and below, the soil is made up of muddy clay. In the geotechnical engineering field, the hard layer is usually used as a bearing layer, like the hard sandwich used in previous engineering practices [11]. The soft soil below the hard layer must be strengthened because strengthening just the soft upper layer cannot meet the requirements for subsidence after the construction of a high-speed railway. As mentioned in the preface, it is easy to shrink the neck and break the pile after the ordinary pile passes through the hard layer but ensuring the quality of the pile body is hard. Though the super-long bored cast-in-place pile can penetrate the hard layer, it is expensive if the whole subgrade is a pile foundation. Therefore, we adopted a new bag-grouting-pile-composite-foundation technique for the construction project. The drilling schematic diagram is shown in Figure 1.

## Engineering Geology Histogram

| Project Name | Engineering Construction Drawing | | Project Type | Soft Soil Roadbed | | | Elevation | | 3.55m |
|---|---|---|---|---|---|---|---|---|---|
| | | | | | | | Start Date | | 2005−03−20 |
| Numbering | JZ-III05-L5 | Loca-tion | DK234+600.00 | Ax-is | X / y | | End Date | | 2005−03−24 |

| Level | Era | Illustrate | 1:200 | Depth | Thick | Elev-ation | Water Level 0.30 | Spec-imen | Test | Bearing Capacity kPa | Remark |
|---|---|---|---|---|---|---|---|---|---|---|---|
| (1) | $Q_4^{al}$ | Clay: yellow, soft plastic, good toughness | | 2 00 | 2 00 | 1 55 | 05-03 | 1 / 0.70-1.00 | | 100 | |
| (2) | $Q_4^{m}$ | Clay: Grey, soft plastic, saturated. | | 8.50 | 6.50 | −4.95 | | 2 / 2.70-3.00 | 12+10 +13(D) 8.50-8.80 | 40 | |
| (3)−2 | $Q_4^{al+pl}$ | Fine breccia: brown, the particle content of 2-6cm accounts for 40~50% of the total mass, the sand content is 10~20%, the rest is cohesive soil, and the part is coarse breccia. | | 16.70 | 8.20 | −13.15 | | 3 / 9.20-9.50 <br> 4 / 16.70 - 17.00 | 15+17 +21(D) 11.50-11.80 <br> 12+11 +10(D) 15.00-15.30 | 300 | |
| (4) | $Q_4^{m}$ | Muddy clay: gray, fluid, uniform. Locally it is silty silty clay. | | 26.20 | 9.50 | −22.65 | | 5 / 17.70 - 18.00 <br> 6 / 22.90 - 23.20 | | 40 | |
| (5) | $Q_4^{al+pl}$ | Fine breccia: brown, medium density, saturated particle size of 2.00~6.00cm accounts for 50~60% of the total mass, and contains about 10% of cohesive soil. | | 32.50 | 6.30 | −28.95 | | 7 / 26.70-27.00 | 17+19 +21(D) 29.00-29.30 | 300 | |
| (5)−2 | | Muddy sticky main: gray, flow plastic, a small fine sand. | | 34.00 | 1.50 | −30.45 | | 8 / 39.00 - 39.30 | | 120 | |

**Figure 1.** Drilling schematic diagram.

## 3. Bag-Grouting-Pile Technology

After ordinary piles penetrate the hard layer, the problems of necking and broken piles often occur. To address this critical issue, the author co-designed this construction method, which was implemented by the author in this railway project [1]. A bag grouting pile is a cylindrical reinforcement made from a combination of geotextile and grouting slurry. The composition is mainly made up of grouting core pipe, geotextile bag, and cement paste. The construction process (shown in Figure 2) begins with a rig drilling to punch holes and to put the cloth-wrapped grouting core pipe into the hole. After that, cement paste is poured in from the top of the core tube into the cloth. This process will generate grouting pressure that will cause the bag to expand from bottom to top.

It should be noted that the cloth is bound with wire. When the cloth is filled with cement slurry, the soil around the hole wall is squeezed. This will cause a slight increase in the size of the pore. When grouting is completed after 20 min, the pile body is grouted twice. When the maximum temperature is lower than 25 °C, the concrete shall be covered and sprinkled for curing within 12 h. In the initial stage, the amount of sprinkling water is increased appropriately. When the seven-day uniaxial compressive strength of the bag pile is greater than 3.5 MPa, curing is stopped. The curing time is not less than 21 days. Finally, a cylindrical object (30–40 cm) is formed. The composite foundation is made up of the bag grouting pile and soil layer to improve the soil-bearing capacity.

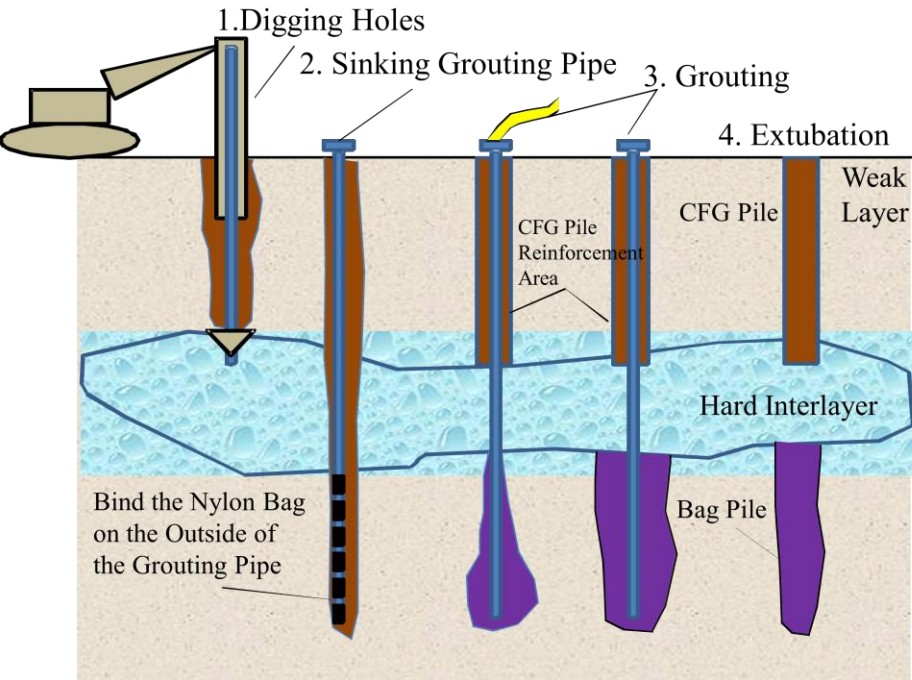

**Figure 2.** Diagram of bag grouting pile.

## 4. Experiment Plan

We conducted a contrast experiment on a selected subgrade section. The bag-grouting-pile foundation was the main test section in the mainline section, and the part of the foundation that was unburied in the field-station section was the auxiliary section. We had different requirements for subsidence in the different areas. The requirements for the mainline and the station sections were less than 15 cm and 30 cm, respectively. The test section met the design requirements and explained the control function of the bag grouting pile.

The main test fracture was reinforced with a gunite mixing pile and bag grouting pile. The upper soil layer was reinforced using the grouting-pile-reinforcement method, with the hard interlayer as the boundary, while the bag-grouting-pile-reinforcement method was used for the soft soil. The diameter of the gunite mixing pile was 0.5 m, the pile distance between the two piles was 1.1 m, and the pile length varied from 7.0–18.0 m. The piles were arranged in an equilateral triangle. The bag grouting pile adopted the rotary-drilling method with an opening diameter of 89 mm, pile diameter of 40 cm, and pile spacing of 1.6 m. The grouting thickness ranged from 4.0 to 19.5 m and was arranged in a square.

The following were observed during the test: layered subsidence of foundation, lateral movement of foundation, surface subsidence, pile, and earth pressure. In order to test the stratified settlement of the foundation, a stratified settlement magnetic ring was buried in the center of the subgrade, and a magnetic ring was arranged every 2 m in the depth direction, with a total of 12 magnetic rings. In order to test the pile earth pressure, three groups of earth-pressure cells (arranged in the subgrade, the center of the left line of the subgrade, and the right shoulder) were buried; each group had three pressure cells (arranged in the center of the pile, between the piles, and the soil centroid between the piles). In order to test the lateral displacement of the foundation, one inclinometer pipe was set at the toe of the slope on both sides of the section, with a depth of 27 m. Figure 3 shows the layout of the test components.

We arranged two observation sections on site, namely DK234+598 and DK234+630. Sectional subsidence observations included ground subsidence observations and deep-seated subsidence observations. The specific observation content is shown in Table 1.

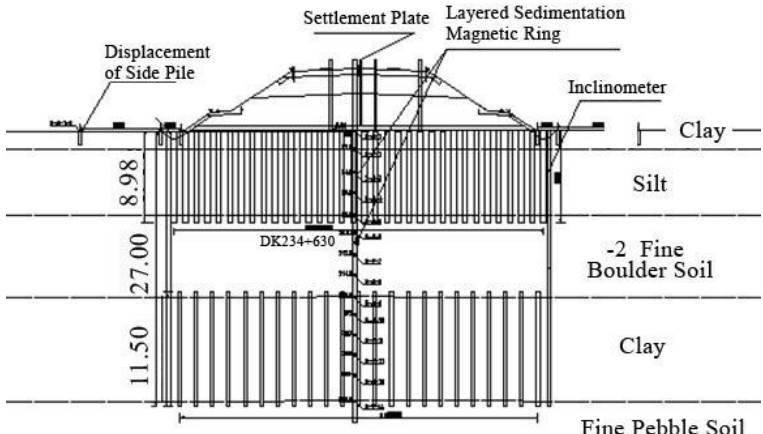

**Figure 3.** Test sites component arrangement.

**Table 1.** Observation record table.

| Type of Sectional Observation | Observation Method | Location | Observation Time | Remark |
|---|---|---|---|---|
| Land subsidence observation | Settling plate | Top of subgrade piles and soil between piles; top of road shoulder piles and soil between piles | 4 Jan 2008–18 Jun 2009 (512 d) | |
| Deep settlement observation | Layered Settlement Method | Middle section of subgrade | 1 Mar 2008–17 Feb 2009 (354 d) | The buried depth of the sinker is 26.5 m, and a layer of magnetic ring is buried every 2 m. |

The frequency of settlement observation was one observation for each layer of filling. When the filling interval was longer, observations were made once every 5 days. After the subgrade was filled in layers to the preloaded design elevation, it was observed once every 5–10 day in the first 2–3 months. After three months, it was observed once every 7–15 day, and after half a year, it was observed once a month. After the observation, the fill-height–time–settlement relationship curve was drawn. The horizontal displacement of the foundation was observed, the test data of the inclinometer tube embedded at the shoulder and the toe of the test section were zero, and the displacement of each section was accumulated to obtain the horizontal displacement curve. Section DK234+600 was tested for pile–soil stress ratio. The soil pressure box was buried in the top of the pile in the center of the roadbed and the soil between the piles. The change in compressive stress in the construction phase was tested and a curve of the time change was drawn. The stress test lasted from 7 January 2008 to 17 June 2009, with a cumulative total of 508 days.

## 5. Test Data Analysis

### 5.1. Subsidence Test Analysis

The Ningbo-Taizhou-Wenzhou Railway has been in operation since 2009. The daily work of railway track inspection reports that the settlement does not exceed the limit. According to China's high-speed-railway-management regulations "Railway Safety Management Regulations", it is not allowed to enter the railway safety range to carry out related settlement surveys.

Figure 4 shows the foundation's subsidence curve. The subsidence curve of the pile top center and the soil center between the piles converges in the middle of the roadbed and road shoulder. The maximum subsidence occurred between piles in the middle of the subgrade, with total subsidence of 152 mm. In the middle of the subgrade, the maximum cumulative subsidence of the pile top was 144 mm. Figure 5 shows the subsidence change in the

section in the auxiliary comparison test, and the maximum subsidence value was 321 mm. The aforementioned data show that the amount of the bag-grouting-pile foundation's accumulated subsidence was less than that of the unburied bag-grouting-pile foundation.

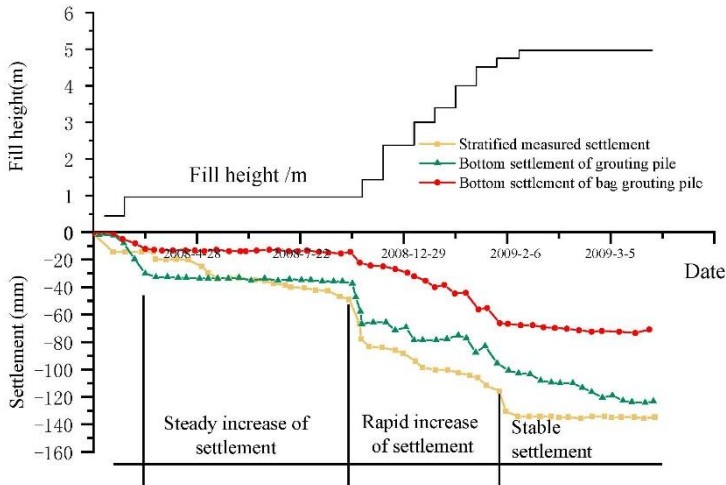

**Figure 4.** Load-time–subsidence curve of bag-pile foundation.

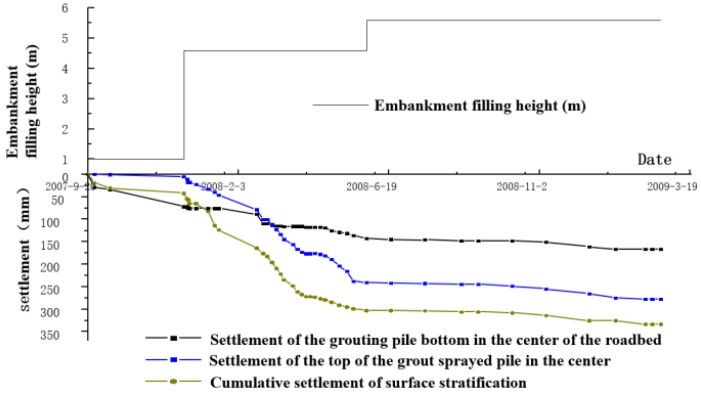

**Figure 5.** Load-time–settlement curve of subgrade center contrast section.

The main methods of subsidence prediction are: Three-point, Hyperbolic, Starfield, Parabola, Asaoka, S-shaped Growth Model, and Gray model. According to the most unfavorable principle, the curve with the maximum value of the measured data was selected as the prediction target curve, and it was fitted with the actual measured data. After comparison, Hyperbolic and Gray Model Gm (1,1) were suitable for settlement prediction of the measured curves. Compared with the measured settlement value before and after fitting, the prediction error was less than 8%, which meets the requirement of statistical error and can ensure the accuracy of the prediction value.

Figure 6 shows the subsidence prediction curve of the bag-grouting-pile test section. The post-stabilization subsidence was 39 mm, which meets the design requirements. After the track was laid on the roadbed, under the action of the load, the section of the bagged pile continued to settle by 39 mm on the original foundation. The overall settlement of the bagged-pile test section was 161 mm. However, the subsidence amount of the auxiliary section of the foundation without embedded bag grouting pile after stabilization was 161 mm, which is above the design requirements of the positive line (<15 cm). Thus, the subsidence test data show that the bag-grouting-pile foundation can strengthen the deep soft-soil foundation interlayered with hard materials on the ballast track.

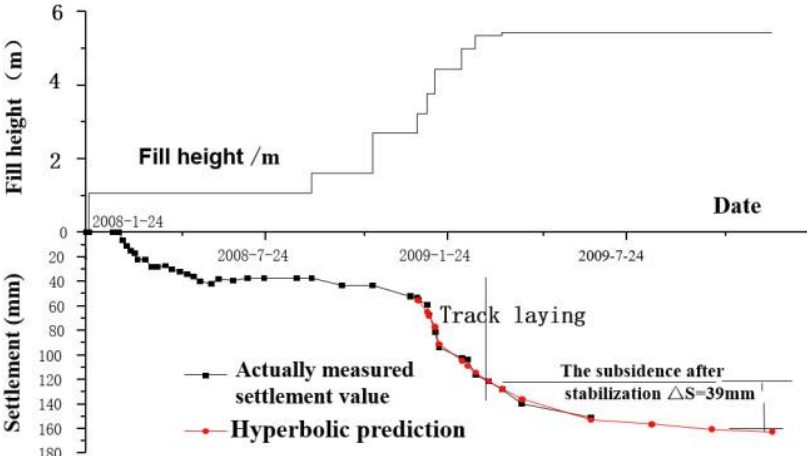

**Figure 6.** Bag-grouting-pile section subsidence curve.

### 5.2. Pile–Soil Load-Sharing Ratio

The load-sharing ratio is a parameter. It is used to show the degree of interaction between piles and soil in the foundation and functions as a design parameter for composite foundations. The pile and the soil between the piles share the load, except when the piles bear most of the load. On the contrary, it is difficult to find the reinforcement effect of composite foundation when the soil between piles bears most of the load. In the test section, the earth-pressure box was placed at the top of the pile to test the earth pressure between the piles. This is to ascertain the load-sharing situation of the bag grouting pile. The test data shown in Figure 7 indicate that increased filling load leads to increased soil pressure on the pile top and between piles. When the fill height increased to 5.62 m, the earth pressure on the pile top and between piles reached its peak and then rebounded slightly. The maximum recorded soil pressure for the basement pile top was 377 kPa, and the maximum between piles was 113.47 kPa.

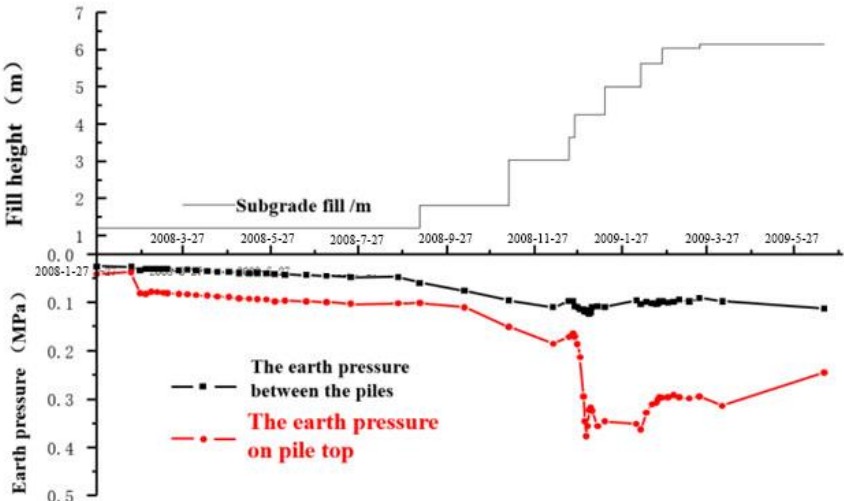

**Figure 7.** Pressure curves of the bag-pile section pile top and the soil between piles.

Figure 8 shows the main section's pile–soil stress ratio. An increased filling load led to an increased pile–soil stress ratio. Afterload stabilization, the pile–soil stress ratio decreased. When the soil between piles was compressed and consolidated, the piles and the soil between piles formed a composite foundation. Under the condition of constant displacement ratio, the soil between piles bore more of the load share, so the pile–soil stress ratio tended to decrease. At the end of the whole process, the pile–soil stress ratio was 2.4.

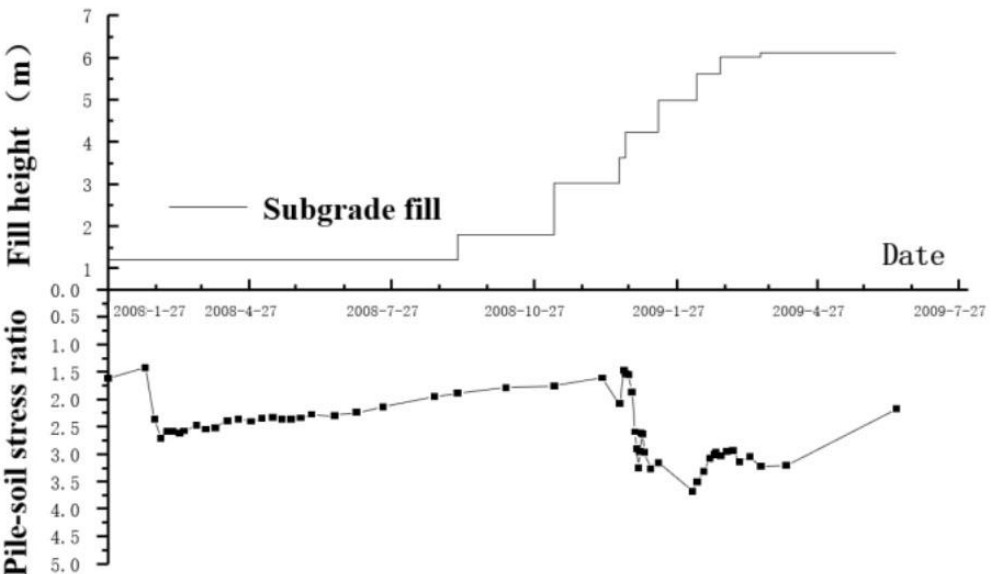

**Figure 8.** Pile–soil stress-ratio curve of bag-pile section.

Compared with the previous research results [12,13], the measured pile–soil stress ratio is similar to that of the flexible composite foundation such as gravel pile. However, the pile–soil stress ratio is significantly lower than that of the tubular pile and CFG pile [14,15]. Therefore, we calculated that the load-sharing ratio of the composite foundation pile of the bag grouting pile was 18.7–47.9% after converting the measured pile–soil stress ratio into the load-sharing ratio. This, from a numerical perspective, shows that the design made reasonable use of pile–soil co-bearing action.

### 5.3. Lateral Displacement

The lateral-displacement distribution and variation rule of the bag-grouting-pile section is shown in Figure 9. After analyzing the angle of the change rate, the change rate of horizontal displacement was less than 2.0 mm/d, which meets the specification requirements for subgrade filling control standards. As for the distribution, the depth-curve direction takes on the shape of an "ear." The curve inflection point is positioned concerning the layered nature of foundation soil and the depth of pile treatment. The first inflection point of the tested section curve is the boundary point between the mixing pile reinforcement area and the underlying layer.

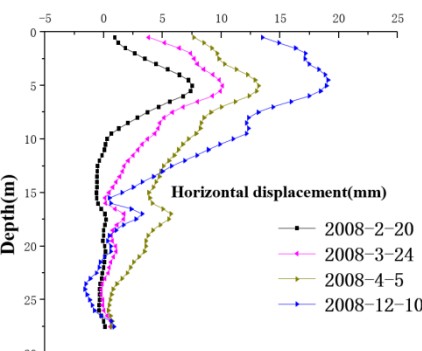

**Figure 9.** Bag-pile section lateral-displacement curve.

Figure 10 shows the distribution curve of the lateral displacement of the auxiliary section along the depth direction. The influence of construction on the lateral-displacement data of the surface without laying out the bag grouting pile has no use value. The data-distribution pattern is similar to that of the bag grouting pile when the hard layer depth

is above 10 m. In the soft-soil range below the hard layer, within the 15~25 m range, the cumulative maximum lateral displacement of the bag grouting pile foundation was 8.9 mm. The displacement change rate shall not exceed 0.3 mm/d at most. Compared with the contrast section, the maximum lateral displacement was 15.4 mm at the same depth, and the maximum displacement change rate was 0.7 mm/d. The displacement and velocity were both greater than that of the bag grouting pile foundation. Considering the compaction effect of the pile, we know that the pile group in the deep soft soil restricts the lateral deformation of the soil. Concerning previous studies [16], we believe that piles generate additional shear stress by reinforcing the soil between piles.

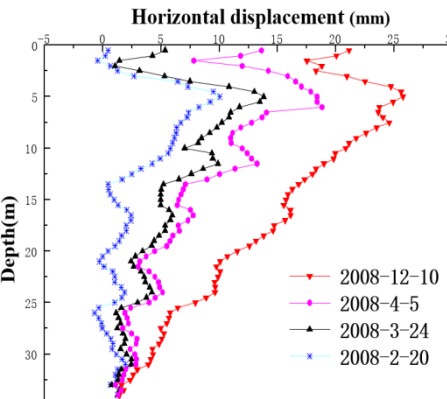

**Figure 10.** Profile horizontal displacement testing data.

## 6. Numerical Calculation of Working Behavior of Bag-Grouting-Pile Foundation

We verified the subsidence-control effect of the bag-grouting-pile foundation and the interaction of pile and soil through a field test. However, the design failed as the cloth's drainage effect could not be tested. In addition, the bag-grouting-pile foundation in the test section can effectively strengthen deep soft-soil foundation with an interlayer of hard materials. But in terms of design, we need to further analyze and verify the possibility of optimization. The bearing capacity of the composite foundation is characterized by the composite modulus and load-sharing ratio. In the compound modulus calculation formula, the main parameters are pile length, pile diameter, and pile spacing. The main material of the cloth bags is nylon fiber. When the cloth bag wraps the pile, it has the function of a plastic drainage board for the surrounding soil. So, the parameters to be optimized are pile length, pile diameter and drainage effect. Therefore, this paper used numerical calculations to discuss these problems. The paper found that under plane strain conditions, the pile element is equal to the strip sheet pile with the width of the pile diameter [17–19]. The equivalent formula of pile modulus is calculated as shown in Equation (1). $E_p$ represents the actual pile modulus. $E_s$ represents the deformation modulus of soil between piles. $S$ represents pile spacing. $D$ represents pile diameter.

$$E_{sp} = \frac{\pi d}{4s} E_p + \left(1 - \frac{\pi d}{4s}\right) E_s \tag{1}$$

### 6.1. Establishment of the Computational Model

DK234+598 of the Ningbo-Taizhou-Wenzhou Railway was the research focus. The railway's foundation was strengthened using the gunite mixing pile and the bag grouting pile. The bag grouting pile was buried in muddy soil. The top ends were respectively embedded with fine pebble soil. The diameter of the grouting pile was 0.5 m. The spacing between the two piles was 1.1 m. The length of the pile was 9.0 m. The diameter of the bag grouting pile was 0.4 m. The spacing between the two piles was 1.6 m. The length of the pile was 11.5 m. According to the static contact and geological drilling data, the cross-section foundation was divided into eight soil layers: covering clay, silt, gravel soil,

silt clay, and so on. The geological conditions were representative, and the embankment fill height was 6.0 m.

According to indoor and outdoor experimental data, each soil layer's physical and mechanical indexes in the model are shown in Table 2. The material parameters of the pile are shown in Table 3.

**Table 2.** Finite-element model materials physical mechanic index.

| The Soil | Thickness | Model | Saturated Unit Weight | Young's Modulus E (MPa) | Poisson's Ratio | Cohesive Force C (KPa) | Angle of Internal Friction ($\varphi$) | The Permeability Coefficient cm/s | |
|---|---|---|---|---|---|---|---|---|---|
| | | | | | | | | Vertical (kv) | Horizontal (kh) |
| (1) Clay-Soft plastic | 1.8 | Moore and Cullen | 17.7 | 2.3 | 0.35 | 12.8 | 9.15 | $4.34 \times 10^{-7}$ | $5.86 \times 10^{-7}$ |
| (2) Silt-plastic flow | 6.4 | Moore and Cullen | 16.8 | 2.93 | 0.45 | 9.12 | 7.13 | $1.46 \times 10^{-8}$ | $5.65 \times 10^{-8}$ |
| (3) fine pebble soil | 8 | Moore and Cullen | 19 | 8.3 | 0.2 | 3 | 30 | $9.45 \times 10^{-5}$ | $4.31 \times 10^{-4}$ |
| (4) Silty clay | 10.2 | Moore and Cullen | 17.7 | 2.92 | 0.45 | 9.74 | 7.85 | $3.15 \times 10^{-6}$ | $3.84 \times 10^{-5}$ |
| (5) fine pebble soil | 6 | Moore and Cullen | 19 | 8.3 | 0.2 | 3 | 30 | $9.45 \times 10^{-5}$ | $4.31 \times 10^{-4}$ |
| (5)-2 Silty clay | 2.2 | Moore and Cullen | 18.2 | 4.37 | 0.45 | 11.6 | 9.42 | $3.15 \times 10^{-6}$ | $3.84 \times 10^{-5}$ |
| (5)-3 Clay-Soft plastic | 7.2 | Moore and Cullen | 17.7 | 2.3 | 0.35 | 12.8 | 9.15 | $4.34 \times 10^{-7}$ | $5.86 \times 10^{-7}$ |
| Roadbed fill | 6.12 | Moore and Cullen | 19 | 3 | 0.25 | 10 | 35 | $4.0 \times 10^{-4}$ | $4.0 \times 10^{-3}$ |

**Table 3.** Pile material.

| Name | Model | EI (kN/m$^3$) | EA (kN/m) | Calculated Thickness (m) | Axial Compression (kN/m$^2$) | Poisson's Ratio |
|---|---|---|---|---|---|---|
| Grouting pile | Linear elasticity | $2.5 \times 10^8$ | $5.0 \times 10^6$ | 0.5 | 24,000 | 0.3 |
| bag grouting pile | Linear elasticity | $3.2 \times 10^8$ | $4.5 \times 10^6$ | 0.4 | 20,000 | 0.3 |

### 6.1.1. Material Parameters

We used the Mohr–Coulomb model to simulate the constitutive relationship of the soil. The material was undrainable in terms of parameter settings. During the filling process of the roadbed, the pore water pressure will increase. The material adopts the effective stress-intensity index and is calculated by the consolidation drainage. PLAXIS uses Young's modulus as the basic stiffness modulus of the Mohr–Coulomb model.

### 6.1.2. Finite-Element Basic Assumptions

The selected subgrade section is based on the plane strain principle, and the basic assumptions are:

(1) The soil is an elastic material, and the Mohr–Coulomb model is used.
(2) The initial stress field of the subgrade and the foundation is generated by the self-weight load.
(3) The contact condition of the pile and soil is partially sliding; the contact interface is simulated by the slip coefficient.
(4) The left and right boundaries of the foundation are impermeable, and the bottom and upper parts are permeable boundaries; the bottom is completely fixed and constrained, and the vertical boundaries on both sides are subject to sliding constraints.

### 6.1.3. Boundary Conditions

When creating an engineering model in the PLAXIS import program, there is no need to display the boundary conditions that define the model. After entering calculation mode, the program automatically creates all boundary conditions of the model based on the following conditions. The default settings are all constraints at the bottom of the model, free at the top, and normal constraints on four sides. At the same time, considering the groundwater level, the stratigraphic conditions, and the boundary conditions of the consolidation analysis to be carried out in the calculation process. Without any additional input, all boundaries are drained. Water can flow out through any boundary. The excess hydrostatic pressure can be dissipated in all directions. There is no free water flow on the left and right sides of the foundation of this model, and the consolidation boundary is closed. The excess hydrostatic pressure of the soft-soil layer can seep into the underlying permeable sand layer (not included in the model), so the bottom boundary should be permeable. The upper boundary is also apparently permeable. We adopt the PLAXIS standard boundary, the vertical boundary on both sides adopts the sliding constraint, and the bottom a fixed constraint. Among them, special attention should be paid to the water level and the boundary conditions of the upcoming consolidation analysis. The excess water pressure can be dissipated in all directions. The superstatic water pressure of the soft-soil layer can seep into the underlying layer, and the bottom and upper boundaries are permeable.

### 6.1.4. Mathematical Model

The model used plane strain 15-node elements. The model was meshed with a total of 1265 elements and a total of 10,769 nodes. We re-encrypted the grids of the pile–soil contact positions and key observation areas. The slab element was used to simulate the pile. The virtual thickness of the plate element was taken as the cross-sectional area of the pile body converted by the area-equivalent method, and the pile length was equal to the actual one. the contact surface element was introduced into the pile–soil interface, and Rinter = 0.65. According to drilling data, eight soil layers were divided. The thickness of the soil layer was set according to the geological data, and the physical and mechanical indexes of each soil layer are shown in Tables 1 and 2. The height of the subgrade filling was 6.12 m, filled in layers. The meshing of the finite-element model is shown in Figure 11.

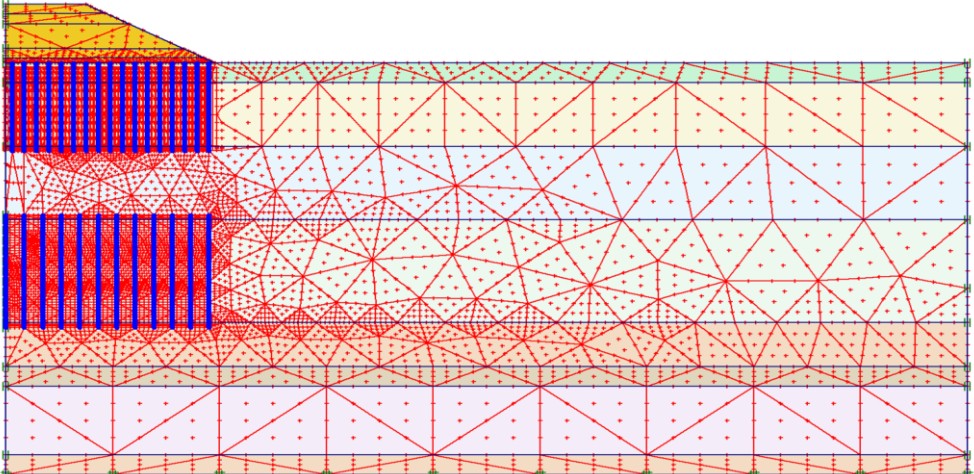

**Figure 11.** Finite-element-model meshing.

### 6.1.5. Included in the Calculations

The finite-element calculation conditions are shown in Table 4.

The location of the observation point was the subgrade centerline and the subgrade soil of different depths under the road shoulder. The purpose of setting up observation points at the top and end of the grout shot pile and bag pile, and the toe of the roadbed

slope is to analyze the law of soil settlement transformation. The simulation calculation included subgrade filling for a total of 514 days. The distribution law of subgrade settlement deformation was obtained by calculation. The data curve of the calculation result of the finite-element-calculation program is shown in Figure 12.

**Table 4.** Finite-element-calculation conditions.

| Phase | Date | Content | Time Interval |
|---|---|---|---|
| 1 | 20 Feb 2008 | Drilling into piles | 10 d |
| 2 | 1 Mar 2008 | Subgrade fill height 0.6 m | 9 d |
| 3 | 9 Mar 2008 | Subgrade fill height 1.2 m | 290 d |
| 4 | 24 Dec 2008 | Subgrade fill height 4.99 m | 11 d |
| 5 | 6 Jan 2009 | Subgrade fill height 5.62 m | 9 d |
| 6 | 15 Jan 2009 | Subgrade fill height 6.02 m | 25 d |
| 7 | 9 Feb 2009 | Subgrade fill height 6.12 m | 8 d |
| 8 | 17 Feb 2009 | Perform consolidation | 150 d |

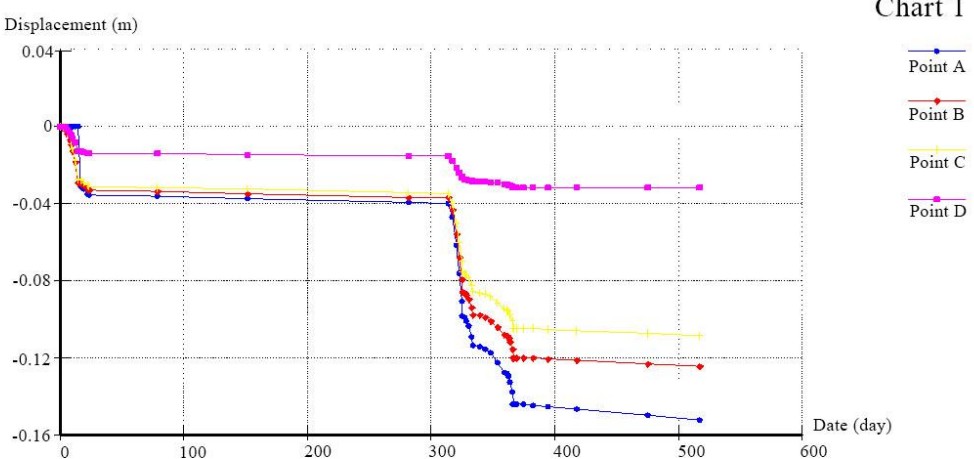

**Figure 12.** Settlement-development curve of numerical calculation.

As of 17 February 2008, the test results are compared as follows: as shown in Table 5.

**Table 5.** Final settlement.

| Section Location | Layout Position | Settlement Plate (mm) | Finite-Element Calculation (mm) |
|---|---|---|---|
| Subgrade Center | Pile top | — | |
| Subgrade Center | Soil between the piles | — | |
| Shoulder | Pile top | 144 | 139 |
| Shoulder | Soil between the piles | 153 | 152 |

*6.2. Force of the Bag Grouting Pile*

The data of axial force and lateral friction resistance were obtained by calculation. Figure 13 shows the distribution of data and the law of development, and shows that the distribution law of axial force and lateral friction resistance of each pile at different positions is the same by analyzing the bag-grouting-pile force. There was obvious negative friction at the pile top and positive friction at the pile bottom, and the neutral point was near the middle of the pile. In addition, there were obvious differences in the axial force and lateral friction resistance of each pile on the subgrade cross-section. This is because

the stress of the pile outside the subgrade was smaller than that of the pile in the center of the subgrade. Therefore, the axial force of the pile was bent in the middle position. After analysis, it is believed that the soil around the pile body had shifted downward.

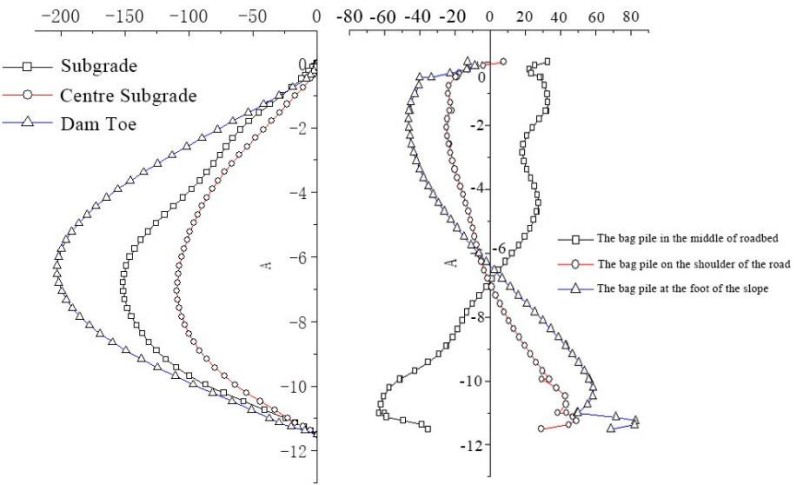

**Figure 13.** Pile bending and friction distribution.

Meanwhile, the pile was subject to downward friction. During the downward transmission of axial force, friction resistance and gravity increment causing the nonlinear increase in axial force along the depth led to the bending of the axial force of the pile. The geotechnical resistance calculation showed that the allowable bearing capacity of the single bag grouting pile was P = 1687 kN. Moreover, the pile in the middle of the roadbed had the maximum axial force of 223 kN. Therefore, the maximum lateral resistance of the pile was 67 kPa. This is far less than the allowable bearing capacity of the bag grouting pile.

### 6.3. Parameter Analysis of the Bag Grouting Pile

Under stable foundation conditions, we set the pile length of the bag grouting pile at different lengths: 9.5 m, 10.5 m, 11.5 m, 12.5 m, 13.5 m, and 15 m. Figure 14 shows the relationship between pile length and the final subsidence. When the minimum pile length was 9.5 m, the final subsidence was 153.69 mm. At 15 m high, the pile passes through the fine pebble soil under the silty soil with a final subsidence of 135.51 mm. The overall subsidence reduction was 11.8%. In addition, the final subsidence amount of the 11.5 m pile was 145.37 mm. After increasing the pile length, the pile passed through the gravel soil layer, and the subsidence decreased by 6.7%. The bag grouting pile is a kind of friction pile. The bottom of the pile is arranged in the thin round gravel soil with a strong bearing capacity. We found that the appropriate increase in its length was conducive to improving its ability to strengthen the foundation. However, the calculation shows that the subsidence-control effect is not obvious with the increase in designed pile length.

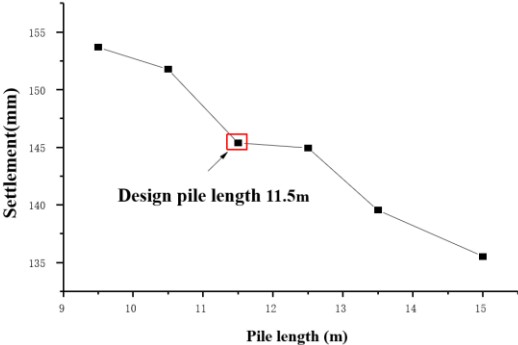

**Figure 14.** Pile-length effect on subsidence curves.

To analyze the influence of pile diameter on the subsidence-control effect, we set the diameter of piles as 0.30 m, 0.35 m, 0.40 m, 0.45 m, 0.50 m, and 0.60 m, respectively. The result is shown in Figure 15. When the diameter of the pile was 0.3 m, the corresponding final subsidence was 148.46 mm. When the pile diameter was 0.6 m, the corresponding final subsidence was 139.51 mm. Therefore, its subsidence decreased by 6.02%. When the diameter of the pile was 0.4 m, its final subsidence as 145.37 mm. After increasing pile diameter to 60 cm, the subsidence decreased by 2.08%. The calculation results show that pile diameter has no obvious influence on subsidence. Next, we set different pile spacing: 1.1 m, 1.3 m, 1.6 m, 1.8 m, 2.0 m, and 2.2 m, respectively. The calculation results are shown in Figure 16.

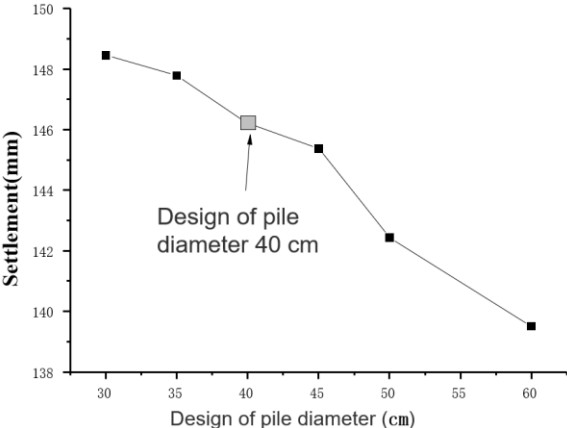

**Figure 15.** Impact of pile diameter on the settlement curve.

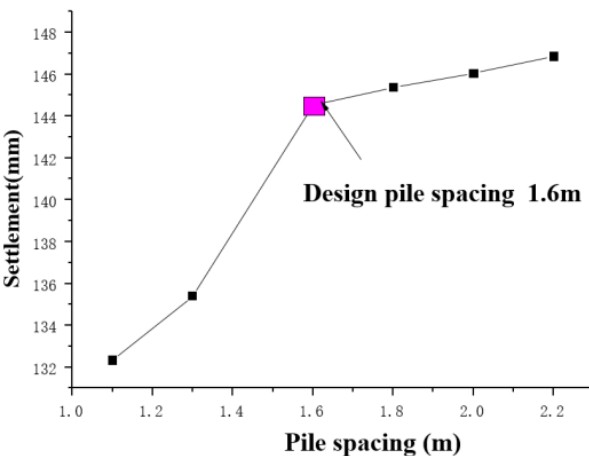

**Figure 16.** Pile-spacing impact on the subsidence curve.

When the pile spacing was 1.1 m, its final subsidence was 132.34 mm. When the pile spacing was increased to 2.2 m, the final subsidence was 146.85 mm. Its subsidence decreased by 9.8%. With the change in pile diameter and pile spacing, the replacement rate will also change. There is no significant difference in the subsidence-control effect. From a cost-saving perspective, we can reduce the cost by appropriately increasing the pile spacing.

So, the change in the length of the bag grouting pile has an obvious effect on the subsidence control. However, we think that the change in the replacement rate is relatively small. This means that it can be used as a friction pile to strengthen the foundation, while changes in pile diameter or spacing have an insignificant impact on subsidence. The pile body with a pile–soil stress ratio between 1–4 was deemed a flexible pile, according to field tests. The calculation shows that the bearing capacity of the pile was not fully utilized. The layout of the square of the pile did not change. We can improve the displacement rate of

the foundation of the bag grouting pile to ensure the subsidence-control standards meet the requirements. For example, we can appropriately increase pile spacing to yield greater bearing capacity and reduce project cost.

### 6.4. Analysis of Drainage Effect of Cloth around the Pile

This treatment technology considers that the outside of the bag grouting pile surrounded by the cloth has a drainage channel in the design. This model used a plate element to simulate piles. We used drainage lines to simulate the cloth drainage in the pile periphery unit. Through calculation, we obtained the dissipation law of excess pore pressure at the monitoring point at a depth of the bag-grouting pile-reinforcement, as shown in Figure 17.

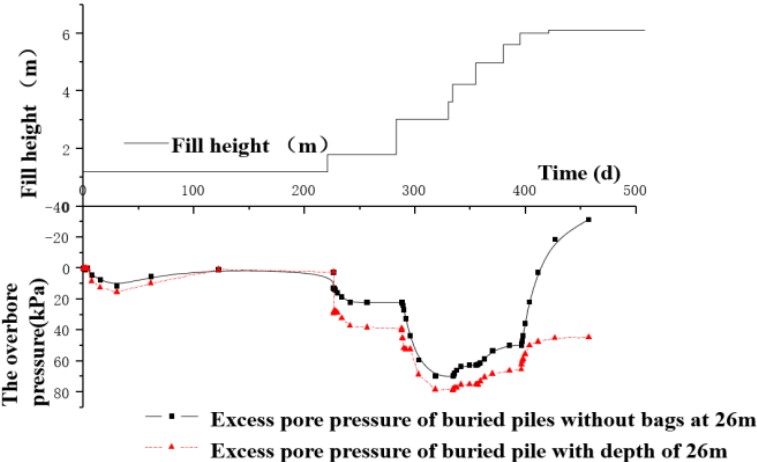

**Figure 17.** Contrast on pore-pressure dissipation of bag-pile foundation.

The overbore pressure significantly increased when the filling loading began. After completion, the overbore pressure became negative when the rate of dissipation of the overbore pressure exceeded the rate of accumulation. This shows that the dissipation rate of pore pressure in the deep soft soil of the foundation was lower than that of the bag-grouting-pile foundation. The conclusion is that the bags around the pile aid in drainage and help dissipate pore pressure and accelerate soft-soil consolidation.

Plaxis 2D: The most suitable algorithm for calculation is automatically selected to ensure the stability of the results. Similar to flac, the convergence of Plaxis has a lot to do with the preprocessing modeling. If there is a slight defect in the modeling, the phenomenon of non-convergence will occur.

### 7. Analysis of Consolidation Law of the Bag Grouting Pile Foundation

The regularity of the bag-grouting-pile-composite-foundation subsidence over time is a characteristic of the consolidation degree. The finite-element calculation results were compared with the measured subsidence curve, the subsidence curve was calculated by traditional consolidation theory, and the subsidence curve of analytic consolidation of composite foundation was drawn. This aids in verifying the rationality of the finite-element calculation of the dissipation law of overbore pressure. The total subsidence was estimated using a formula. We used the composite-modulus method to calculate the subsidence of the grouting-pile-reinforcement area. The stress of the embankment base was transferred to the lower end of the grouting pile, and the subsidence of part of the gravel-soil layer was calculated by the comprehensive layered method, mainly to calculate its rapid subsidence. In the second layer, we used the composite-modulus method to calculate the subsidence of the grout-shotcrete-pile-reinforcement area. Finally, the pressure-diffusion method [15] was used to calculate the pile bottom stress superposition. The final subsidence of the bag-grouting-pile foundation is shown in Table 6.

**Table 6.** Subsidence calculation results.

| Depth of Foundation Soil | Settlement Estimation (mm) |
|---|---|
| Compression capacity of grouting and jet pile reinforcement area (thickness: 9 m) | 31 |
| First lower bed (thickness: 6.5 m) | 139 |
| Compression capacity of the bag grouting pile reinforcement area (thickness: 11 m) | 18 |
| Second lower bed (thickness: 6 m) | 35 |
| The total subsidence | 222 |

As shown in Table 3, the subsidence estimates the compression of the bag grouting pile reinforcement area at 18 mm, and the accumulated compression of the lower soil mass at 34.1 mm. We believe that the total estimated subsidence at this location is 43.7 mm. We obtained the subsidence of 31.6 mm and 22.41 mm through field tests and finite-element calculation, respectively. To obtain the consolidation subsidence curve, the consolidation degree is calculated using Terzaghi's one-dimensional consolidation theory and the improved Shun Takagi interface method:

$$\overline{U}_s = 1 - \frac{8}{\pi^2}[e^{-\frac{\pi^2}{4}T_{vs}} + \frac{1}{9}e^{-\frac{\pi^2}{4}T_{vs}} + \cdots]$$

(2)

$\overline{U}_s$ represents the average degree of consolidation. $\overline{U}_s$ represents the time factor. This is equal to $\beta_s t / H^2$. $H$ represents the maximum drainage distance. $\beta_s$ represents the consolidation coefficient. According to the permeability coefficient $ks$, the cross-sectional area of pile–soil $A_P A_S$, volume compression coefficient $m_{vs}$, and water weight, the formula can be obtained $\beta_s = [(k_s A_p)/\alpha^2]/[m_{vs}\gamma_w(A_p + A_s)]$. Where $\alpha$ represents the ratio of pile diameter to horizontal influence area.

We use the improved Shun Takagi interface method to calculate the average consolidation degree of the foundation under stepwise loading. This formula is as follows:

$$U_t = \sum_{i=1}^{n} \frac{q_i}{\sum \Delta p}[(T_i - T_{i-1}) - \frac{\alpha}{\beta}\left(e^{\beta T_i} - e^{\beta T_{i-1}}\right)e^{-\beta t}]$$

(3)

$U_t$—$t$ represents the average consolidation degree of the foundation after the constant-speed loading of the time multiple loads (%). $Q_i$ represents the average loading rate of a given load (kPa/d). $T_{i-1}$, $T_i$ represents the start and end times (from zero) of the loading of a class I load, respectively. When calculating the degree of consolidation at a certain moment in the loading process of a certain first-order load, $T_i$ is changed to $T$.

$\alpha$ and $\beta$ represent the drainage consolidation correlation coefficient. We chose the pile bottom soil in the bagged-pile-reinforcement area in the center of the roadbed as the comparison monitoring point, and the test curve and the finite-element curve were obtained. The consolidation subsidence calculation curve can be obtained by formula $S_t = U_t \times S$.

As shown in Figure 18, we concluded by comparing the subsidence and consolidation curves. The finite-element-calculation results strictly correspond to the actual filling process. Overall, the regularity remains consistent. In contrast, Terzaghi's one-dimensional consolidation equation does not consider the loading process. However, the improved Shun Takagi method takes into account the loading time and loading rate. Therefore, the law of the comparative subsidence and consolidation curve is more consistent with the measured curve.

On the whole, when the depth was 26 m, the value obtained by finite-element calculation was too small, b this law has a high degree of agreement. Therefore, this value can be used as the basis for describing the consolidation deformation law of the bag-grouting-pile foundation, and it also verifies the calculated dissipation law of excess pore pressure.

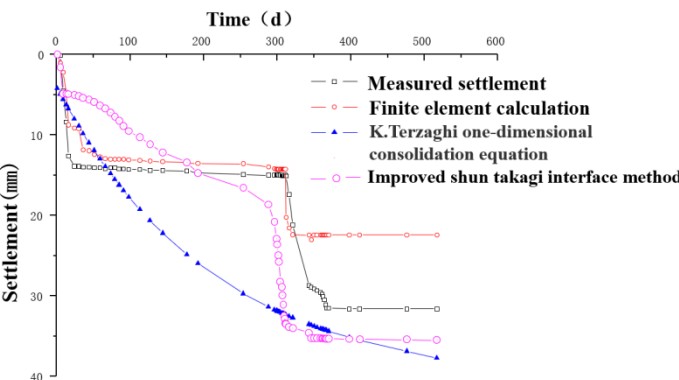

**Figure 18.** Subsidence calculation contrast curve.

## 8. Conclusions and Suggestion

To solve the subsidence control of thick, soft ground with a hard layer, the Ministry of Railways adopted the new foundation-treatment technology of the bag grouting pile in the test section of Ningbo-Taizhou-Wenzhou Railway. The technology's feasibility was verified through numerical calculations. The following conclusions were made:

(1) The settlement rate and lateral-displacement-change rate of the bag grouting pile meet the code requirements. After completing the construction, the settlement did not exceed the design standard for the main roadbed. This indicates that the bag grouting pile can be used in the bag-grouting-pile reinforcement of a deep soft-soil foundation with a hard interlayer.

(2) As for the depth, the distribution and variation of lateral displacement show that the bag grouting pile had a restrictive effect on the deformation of deep soft soil. The dissipation of excess pore pressure was faster than that of the general foundation through calculation. This helps accelerate the consolidation of soft soil under the hard layer and verifies the function of the drainage channel considering the bag around the pile in the design.

(3) Through field tests, the pile body with a pile–soil stress ratio between 1–4 was deemed a flexible pile. The pile's calculated axial force and lateral resistance were 223 kN and 67 kPa, which are far less than the allowable bearing capacity. It was also found that pile spacing had less influence on settlement control than pile length diameter. Therefore, we can choose to increase the distance between bagging piles to reduce the project cost.

(4) It is suggested that in future research, the factors affecting the bearing capacity of bagged piles can be analyzed in combination with model tests; the application properties of bagged grouting piles in karst, beach silt and other soft foundations can be strengthened.

**Author Contributions:** Conceptualization, F.C.; Data curation, S.Z., Q.Z., P.L. and Y.C.; Formal analysis, S.Z. and Q.Z.; Funding acquisition, S.Z., J.L. and F.C.; Investigation, P.L.; Methodology, S.Z. and F.C.; Project administration, Q.Z.; Resources, J.L., P.L. and Y.C.; Software, F.C.; Supervision, Q.Z., P.L. and Y.C.; Validation, J.L. and P.L.; Writing—original draft, S.Z.; Writing—review & editing, J.L. All authors have read and agreed to the published version of the manuscript.

**Funding:** This study was supported by the key R&D projects in Shandong Province under grant number 2019GSF111008 and the Natural Science Foundation of Shandong Province (grant number ZR2020QE260) and the Shandong Provincial Department of Transportation Science and Technology Plan Project under grant number 2021B117, the National Natural Science Foundation of China (grant numbers 52178182 and 52108262).

**Institutional Review Board Statement:** Not applicable.

**Informed Consent Statement:** Not applicable.

**Data Availability Statement:** The data that support the findings of this study are available from the corresponding author.

**Acknowledgments:** This work is supported by the Found of Shandong Key Technology Research and Development Program (2019GSF111008), the Funding Supported by the Natural Science Foundation of Shandong Province (grant number ZR2020QE260), Science and technology plan of Shandong Provincial Department of transportation (2021B117), the National Natural Science Foundation of China (grant numbers 52178182 and 52108262). The authors gratefully acknowledge their financial support.

**Conflicts of Interest:** The authors declare no conflict of interest. The funders had no role in the design of the study; in the collection, analyses, or interpretation of data; in the writing of the manuscript, or in the decision to publish the results.

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
