# Peer review of "Study of Bag Grouting Pile Reinforcing Deep Soft-Soil Foundation with an Interlayer of Hard Materials on High-Speed Railway Ballast Track"

_applsci, doi:10.3390/app12094662_

Round 1

Reviewer 1 Report

This paper presents the a study of bag grouting pile reinforcement deep soft soil foundation with an interlayered of hard materials on high-speed railway ballast track. The paper presents an interesting engineering problem. However, authors need to address the following comments before accepting it for publication.

  1. First, the authors should distinguish the contribution of this work from that of the existing similar works. The last paragraph of the introduction should be strengthened to include the authors' significant contribution and novelty of the work.

  1. The Abstract is very poorly written. Need to be rewritten including the methodology in the test before listing the results of the study.

  1. The references in the main text should be in square bracket [ ] following the MDPI and the journal referencing format

  1. Line 250-252 and associated equation numbering should be corrected. Same applies for all equations in the text.

  1. Line 278, the authors made a statement that they adopted the PLAXIS standard boundary. Please add explanation on why you decided this. Also, included the limitation and all assumptions used in the Finite Element modeling.

  1. Also, in the FE modeling, provide a mesh convergence study.

  1. The conclusion section can be improved. Also, add a paragraph on your recommendation on future studies.

Reviewer 2 Report

This work validated the performance of bag grouting pile on high-speed railway ballast track.  The study is based on measuring the settlement from a railway project.  The reviewer suggests publication with minor revision.  

  1. Page 2 line 61-78 Suggest Add schematic picture on the drilling soil
  2. Page 3 Line 81. Please clarify whether the author has “designed a method of bag grouting into piles” or this is a method is first developed by others [1] and is implemented by the author in the railway project. If the method is not first invented by the author, please add a citation. 
  3. Page 3 figure 1. Please make improvement on the picture quality. 
  4. Page 4 table 1. It looks like the author is comparing the 1)bag grouting pile and 2)gunite mixing pile.  Is it so?  It looks there is no difference in the listed compared area. Consider remove the table and put it into main text.
  5. Page 4 Fig2. Please improve the quality of the picture.   
  6. Page 4-7 Fig 3-7. Please be very clear on the figure title whether the figure is about the “ bag grouting pile” or the control/contrast/comparison section of “gunite mixing pile”
  7. Did the author tried to simulate the conventional “gunite mixing pile” technique?

Round 2

Reviewer 1 Report

Point 1: First, the authors should distinguish the contribution of this work from that of the existing similar works. The last paragraph of the introduction should be strengthened to include the authors' significant contribution and novelty of the work.

It was not fully answered. The reviewer didn't ask the funding information. State again the novelty of your work. What makes this work different from similar research??

Point 5:  Also, included the limitation and all assumptions used in the Finite Element modeling.

Not answered!!

Please consider revising your manuscript.
